# MoE-Infinity: Efficient MoE Inference on Personal Machines with Sparsity-Aware Expert Cache

## Abstract

This paper presents MoE-Infinity, an efficient MoE inference system designed for personal machines with limited GPU memory capacity. The key idea for MoE-Infinity is that on personal machines, which are often single-user environments, MoE-based LLMs typically operate with a batch size of one. In this setting, MoE models exhibit a high degree of activation sparsity, meaning a small number of experts are frequently reused in generating tokens during the decode phase. Leveraging this idea, we design a sparsity-aware expert cache, which can trace the sparse activation of experts during inference and carefully select the trace that represents the sparsity pattern. By analyzing these selected traces, MoE-Infinity guides the replacement and prefetching of the expert cache, providing 2.7–13.7× per-token latency improvements over numerous state-of-the-art systems, including vLLM, Ollama, DeepSpeed and BrainStorm across various MoE models (DeepSeek and Mixtral) when handling different LLM tasks.

**Code**: https://anonymous.4open.sc ience/r/MoE-Infinity-220D/

## 1. Introduction

Mixture-of-Expert (MoE) architectures can significantly improve the efficiency of Large Language Models (LLMs). In an MoE model, the majority of parameters belong to the MoE layers, where numerous experts are connected to a router that dynamically routes input tokens to different experts for processing. During inference, each input token typically activates only a small subset of experts, significantly reducing compute and memory bandwidth demands compared to fully activated counterparts, where all experts must be utilized.

Given its efficiency, LLMs with MoE architectures are increasingly favoured for local deployment on personal machines. Unlike cloud servers, personal machines often have only a single consumer-grade GPU with limited memory (typically 24–48GB). To serve large MoE models—some

| Systems | GPU idle time | TPOT Avg. | TPOT Tail |
|---|---|---|---|
| DeepSpeed | 513ms | 737ms | 803ms |
| Mixtral-Offloading | 754ms | 1250ms | 1530ms |
| Llama.cpp | 2073ms | 2590ms | 2599ms |
| vLLM | 254ms | 485ms | 493ms |
| **MoE-Infinity** | **51ms** | **173ms** | **189ms** |

Table 1: MoE generation latency for DeepSeek-V2-Lite as time-per-output-token (TPOT) on single NVIDIA-A5000-24GB through PCIe4.0 (24GB/s). MoE-Infinity achieves 2.7–13.7× latency reduction. Other systems incurs high latency due to high volume but inaccurate prefetch blocking GPU in addition to low GPU cache hit rate.

exceeding 100GB (Jiang et al., 2024; The Mosaic Research Team; XAI, 2024), such as DeepSeek-MoE (DeepSeek-AI, 2024) with 236 billion parameters—the LLM inference system often relies on offloading (Aminabadi et al., 2022). This means the full MoE model resides in host memory, and only the activated experts (i.e., those receiving tokens for processing) are fetched into the GPU when needed.

Many state-of-the-art inference systems now support offloading, but they often suffer from slow performance. Our trace analysis reveals that the primary cause is poor cache design when fetching experts into GPUs. Most inference engines (e.g., DeepSpeed (Aminabadi et al., 2022), Mixtral-Offloading (Eliseev & Mazur, 2023)) use prediction-based methods to manage their expert cache within GPUs. These methods primarily analyze the execution order of experts in the computational graph (e.g., prioritizing experts in the next immediate layer). While prediction-based methods work well for fully activated dense models, they fail to account for the sparse activation of experts and assume all required experts must be fetched into GPUs, leading to substantial I/O bottlenecks on the PCIe bus. As a result, they suffer from high GPU idle time and thus poor Time Per Output Token (TPOT)—a critical metric for LLM serving, shown in Table 1. In fact, inaccurate predictions can degrade performance even further than on-demand fetching approaches like vLLM (Kwon et al., 2023), which experience less I/O contention but still underutilize GPUs.

Recently, BrainStorm (Cui et al., 2023) was designed for caching parameters of dynamic neural networks. However,

it can only handle dynamic patterns caused by control operators (e.g., if-else) in neural networks in classification tasks, rather than routers in MoE architectures during the auto-regressive LLM decoding process. As a result, it still produces highly inaccurate predictions, performing worse than on-demand vLLM – 934ms TPOT (avg.) in BrainStorm versus 485ms TPOT in vLLM.

We aim to design an effective caching strategy for MoE inference on personal machines. Our key idea is that MoE inference on personal devices typically operates with a batch size of one. Unlike cloud-based environments that batch multiple user requests using techniques like continuous batching (Yu et al., 2022), personal machines are single-user environments and often process a single prompt at a time when running locally deployed LLM services (e.g., ChatBot). During decoding, an MoE model activates only a small subset of experts per token, meaning a small cache is effective (i.e., cache size is comparable to the memory size of a GPU). To optimize caching, we could further develop prediction methods that account for sparse expert activation patterns and identify which experts are most likely to be reused during decoding. This reuse pattern stems from how experts are trained—experts with similar expertise are often reactivated within the same context and prompt, with further analysis and evidence provided in Section 4.4.

Building on this key idea, we introduce MoE-INFINITY, a new MoE inference system for personal machines. The core innovation of MoE-INFINITY is a Sparsity-Aware Expert Cache which leads to the following contributions:

**Contribution 1.** We conduct extensive trace analysis on MoE models and provide novel evidence supporting the feasibility of an effective cache design for personal deployment scenarios. With a batch size of one, we find that the memory available on a consumer-grade GPU is often sufficient to store the most frequently used experts during the decoding phase of an LLM, even with long-context inputs, meaning that maintaining a cache for frequently used experts could be effective in such a case. Additionally, we observe that experts exhibit skewed reuse patterns when continuously decoding tokens within a single request (i.e., a prompt). However, this reuse pattern is only present at the request level; after processing multiple requests, the skew disappears, and all experts tend to be activated uniformly.

**Contribution 2.** We formulate the problem of online prediction for the skewed reuse patterns of experts. By statistically modeling these patterns, we conduct extensive analyses to identify prediction methods that offer robust, real-time performance. Based on this analysis, we develop an expert activation prediction method that traces the sparse activation of experts and carefully selects traces that can guide future predictions. This method is then integrated into a high-performance expert cache. Additionally, we analyze the

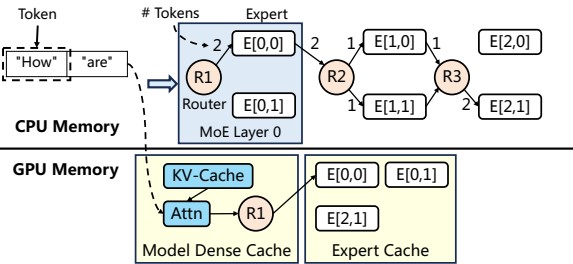

Figure 1: MoE inference on GPU with full model offloaded onto CPU memory. E[0,1] refers to an expert module at layer 0 with index 1.

cache's memory requirements and worst-case performance.

**Contribution 3.** We compare MoE-INFINITY against several advanced inference systems, including vLLM, Llama.cpp, Mixtral-Offloading, DeepSpeed, and Brain-Storm, across various MoE models such as DeepSeek-MoE, Arctic-MoE, Mixtral-MoE, Meta NLLB-MoE, and Google Switch-MoE. Evaluation results show that MoE-INFINITY effectively utilizes its PEC to achieve high cache performance, resulting in 2.7–13.7× improvement in performance on a commodity GPU.

MoE-INFINITY is open-sourced on GitHub. It unlocks local deployment of large MoE models for a vast number of personal machines and is rapidly gaining attention from both industry and academia.

## 2. Background: MoE Inference and Offloading

We provide the necessary background to understand an MoE inference running with offloading on a personal machine. As shown in Figure 1, a copy of expert parameters stays in Host memory, while densely activated parameters (i.e., attention and KV-cache) are cached in GPU memory without eviction. Each MoE layer consists of a router and a group of experts, which are Feed Forward Networks (FFNs). The router assigns each token to specific experts. MoE models can vary in configuration. For example, Mixtral-8x7B has 8 experts per layer, with each expert managing 0.15B parameters (340MB), and the router selects 2 experts per token. Switch-128x0.2B, while similar in total model size, features 128 experts per layer, each expert managing 33MB parameters, with the router selecting 1 expert per token.

The MoE model example processes one prompt (referred to as request). To process these prompts, the model first enters a prefilling phase. Then, it moves into a decoding phase, which iteratively generates outputs. This interaction with GPU is illustrated in Figure 1, after attention and router decides the experts to be activated, the offloading systems look into *expert buffer* to find available parameters. If parameters

are not availble, the system needs to fetch them into the buffer on-demand. To improve latency performance, the offloading system often has an *expert activation predictor*, which facilitates the router decision and prompts to decide the expert to prefetch, overlapping with the computaion of attention and experts in GPU.

## 3. Related Work

We describe all related work regarding offloading and LLM inference. Offloading has been extensively studied in the context of dense neural networks, and systems like Flex-Gen (Sheng et al., 2023), DeepPlan (Jeong et al., 2023), and SwapAdvisor (Huang et al., 2020) do not natively support MoE deployment. More generic offloading systems such as DeepSpeed-Inference and HuggingFace-Accelerate support MoE models but simply treat MoE layers as dense layers, failing to account for the conditional, sparse activation of experts during inference.

Some recent memory swapping systems, such as Sentinel (Ren et al., 2021) and DeepUM (Jung et al., 2023), can trace memory access in deep learning models, but they do not trace at the expert level. This limitation results in a high fault rate and negatively impacts performance.

Several inference systems have recently added support for MoE models. Mixtral-Offload (Eliseev & Mazur, 2023), Ollama (Llama.cpp) (Ollama, 2024), vLLM(Kwon et al., 2023), and BrainStorm (Cui et al., 2023) can support MoE but still suffer from poor cache performance (in Table 1).

InfiniGen (Lee et al., 2024) and TensorRT-LLM (NVIDIA, 2024) are designed for multi-GPU environments, making them more suitable for cloud servers rather than personal machines. As a result, they lack the required optimized cache designs, leading to worse performance compared to vLLM, Ollama, and Mixtral-Offload.

## 4. Sparsity-Aware Expert Cache

In this section, we first explain why a small expert cache suffices for accelerating offloading in LLM decoding. We then identify the sparse activation pattern needed to optimize cache performance and make online predictions for better cache replacement. Next, we formulate the online prediction problem and highlight the limitations of conventional tracing methods. Finally, we introduce our request-level sparse activation tracing method, leveraging the sparsity trace to enhance expert cache efficiency.

### 4.1. Why expert cache could be effective

A key requirement for an expert cache to be effective in LLM decoding is that only a small subset of experts is used during each iteration of the decoding phase. This

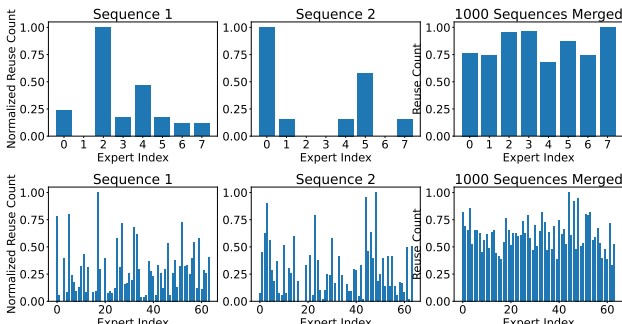

Figure 2: Expert reuse count over decoding iterations for two sample sequences and merged over 1000 sequences. Darker colour means higher reuse normalized. Sampled from last layer of Mixtral-8x7B (top, 20 decoding iterations) and DeepSeek-V2-Lite (bottom, 256 decoding iterations).

indicates a small working set (i.e., maximum capacity) that can fit within the limited memory of a GPU. Otherwise, an excessively large expert cache exceeding GPU capacity would trigger excessive I/O bandwidth usage, slowing down inference due to frequent offloading.

To estimate the size of this working set, we conduct an extensive trace study across widely used MoE models for various LLM inference tasks, with key findings highlighted below. For MoE models with around 100 experts (e.g., DeepSeek, QWen-MoE, NLLB, and Switch-MoE), fewer than 5% of experts are repeatedly activated when decoding tokens for a single request. Even for MoE models with fewer experts (e.g., Mixtral), we observe only 25% activation per request. These results indicate that experts are sufficiently trained to specialize in handling different types of requests.

### 4.2. Why expert cache must be sparsity-aware

A key challenge in expert caching is determining which expert to replace when the cache is full. The optimal strategy depends on predicting which expert is least likely to be activated in the near future, thereby improving the cache hit ratio. A straightforward approach is to track expert activation frequency over time, as implemented in BrainStorm (Cui et al., 2023) and advanced trace-based memory swapping systems like DeepUM (Jung et al., 2023).

However, we find this approach insufficient for expert caching, as it fails to account for the sparse activation patterns of individual requests. Figure 2 illustrates this with two sampled LLM requests. In Mixtral's last layer, Sequence 1 shows Expert 2 being reused over 15 times—7 times more than any other expert—while in Sequence 2, Expert 0 and Expert 5 exhibit the highest reuse counts. Similarly, in DeepSeek, despite having 256 decoding iterations, expert activation remains sparse. In Sequence 2, Experts 48, 44, 23, and 3 exhibit higher reuse than others.

While expert reuse patterns are skewed within single re-

quests, they become more uniform across multiple requests. Analyzing over 1000 sequences, we observe that reuse counts even out over time. With this uniform distribution of expert usage, the expert cache will fail to find which experts are more likely to be reused in a decoding phase, compromising cache performance.

| Model | Switch | | | NLLB | | | Arctic | Mixtral |
|---|---|---|---|---|---|---|---|---|
| | BigBench | Flan | MMLU | BigBench | Flan | MMLU | MMLU | BigBench |
| # Groups | 10 | 15 | 19 | 15 | 30 | 18 | 20 | 28 |
| Nor. Max. Group | 0.478 | 0.390 | 0.252 | 0.125 | 0.062 | 0.145 | 0.100 | 0.121 |

Table 2: Number of group by elbow-point in K-means. The group size is normalized by the total sequence number.

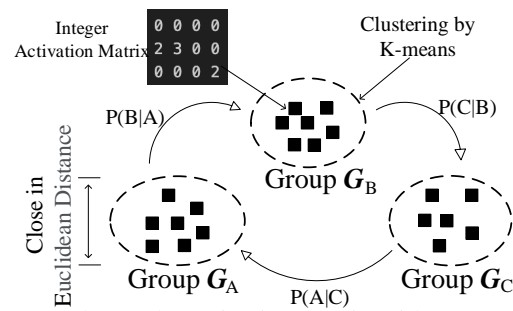

Figure 3: Cluster the activation matrix with K-means, the matrix within the same group has similar value. The activation state is modelled by a Markov Chain.

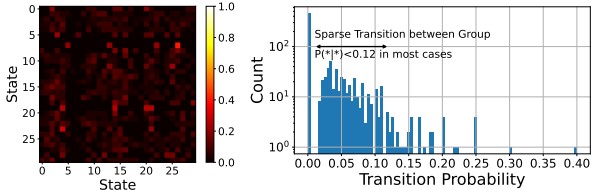

Figure 4: Markov Chain transition matrix for all groups (left) and probability of one group (right) of Arctic with MMLU.

### 4.3. Formulation of predicting activation during decode

We formulate the problem of making online prediction for expert activation during an LLM decoding process. We define the input to the problem as the *Expert Activation Matrix* (EAM). For a model with $L$ MoE layers and $E$ experts per layer, an EAM $M$ is an $L \times E$ matrix where $M[i][j] \in \mathbb{Z}$ is the number of tokens routed to expert $E[i, j]$, i.e., the expert with index $j$ at layer $i$. Each element in the matrix accounts for the number of tokens processed by the expert. Given $n$ tokens has been proceeded by the MoE model, we have $M[i][j] \in \{0, \ldots, n\} \ \forall i, j$ and $\sum_j M[i][j] = n \ \forall i$, as each MoE layer needs to process tokens received by the model.

The online prediction problem uses EAM to find out the activation likelihood for future experts and reuse likelihood for all experts. The online prediction is triggered after knowing the routing decisions in each MoE layer $i$. The predictor provides a predicted EAM (pEAM) with each entry for either reuse or activation. The pEAM is also a $L \times E$ matrix with likelihood as element. A zero in the matrix means that an expert is predicted to be inactive or will not be reused during decoding phase.

The EAM is built at request-level (rEAM) with updates from each iteration (iEAM) as follows: *(1) Iteration-level EAM* (**iEAM**), A iteration-level EAM keeps a trace for each sequence in each forward pass of the model. Formally, the first iteration is the prefilling phase of the model, with number of tokens $n$ equals to the sequence length. While, the rest iterations belongs to decoding phase with $n = 1$. Each iteration-level EAM is updated as per layer of inference. Given the current MoE layer is $l$, $\sum_j M[i][j] = 0 \ \forall j > l$ and $\sum_j M[i][j] = n \ \forall j \leq l$. *(2) Request-level EAM* (**rEAM**), A request-level EAM accumulates the counts of per iteration EAM. Formally, given the total number of tokens across all iterations as $r$, we have $\sum_j M[i][j] = r \ \forall i$. The request-level EAM are traced for prefilling and decoding separately. In prefilling, $r$ is the number of tokens in a prompt, while in decoding, $r$ is the number of output tokens. At the end of each iteration, the iteration-level EAM is added to an accumulated request-level EAM, which tracks the frequency of expert usage since the beginning of the current request.

### 4.4. Intuitions for an effective prediction method

Accurate prediction of expert activation during each iteration is needed for prefetching, and throughout decoding

for caching. Our key observation is that **expert activation prediction is only made possible through matching observed activation patterns among groups of expert**. We validate this in two folds: (i) Existence of expert activation groups, (ii) Transition between different groups is hard to predict. By prediction we consider few existing methods in the literature (Appendix A), none of the existing work fully meets the SGR model for MoE.

First, the existence of strong similarity between rEAM makes the activation predictable based on known patterns. We utilize the selective activations captured in the element of rEAMs, such that each EAM represents a group activation pattern. We apply K-means clustering on a set of rEAMs, where each EAM is obtained per sequence. K-means is used to ensure matrices within the same group exhibit significant Euclidean similarity, closely aligning with the activations. We record the elbow point of the group number and the relative size of the maximum group in Table 2. The ability to form significant cluster means that we can use one EAM to infer others within each clustered activation group. The number of of group activation is relatively small in Table 2, where there are only 10 to 30 groups of activation patterns given 1000 samples for each dataset. The theoretical upper bound of the number of groups is illustrated in Appendix B.1.

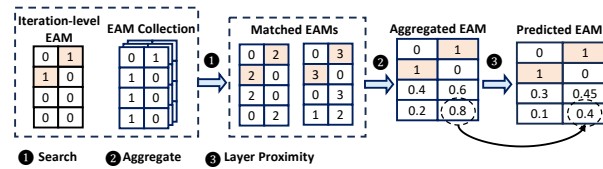

Figure 5: Example of computing activation likelihood.

Second, the prediction of group transition is hard since all group have sparse activation pattern at request-level, i.e., a small activation probability. Except 'Switch', the other models do not show significant major group, meaning that all groups have low activation frequency. The low activation frequency indicates that expert reuse is limited to request-level EAMs, while a single reuse pattern does not apply to the majority of inputs.

We further show that the transition probability between groups is low in Figure 4, meaning using existing group information to extrapolate other groups is infeasible. The highest probability is around 0.3, with most of the lower than 0.12. We also consider longer input sequence for prediction but the conditional probability is still small in most case. Therefore statistically we do not find significant activation pattern for predicting the transition between different groups. This excludes post-hoc learning-based predictions, as data-shift from training set, it is essentially predict the transit between groups.

**Takeaway.** On request-level, the activated experts can be grouped into a limited number of EAMs. However, the transition between EAMs under different requests is unpredictable. Therefore, a reasonable prediction of expert activation is leveraging group matching for new requests.

### 4.5. Request-level sparse activation tracing

We thus must trace the sparse activation of experts at the request level and our tracing must reflect the entire group of experts. For this purpose, we have designed a novel data structure termed the *Expert Activation Matrix Collection* (EAMC), which acts as a trace for keeping historical request-level EAMs online. As the system has processed an incoming request, it compares the request-level EAM with those recently stored in the EAMC. A matching prior EAM can then facilitate more effective prefetching and caching decisions by the serving system. To determine if two EAMs match, we use the following method: each EAM is flattened into a vector, and the cosine distance between these vectors is calculated. Within the EAMC, the most closely matching prior EAM to a current EAM is the one with the smallest cosine distance. The distance measure considers the following: (i) need for the relative frequency of expert activation as sequences has varying length and number of iterations are indeterministic, and (ii) need to handle sparse vectors as expert activations are sparse and skewed, also matching

experts with high activation frequency is beneficial than ones with low frequency.

Given the EAM collection, we define the activation likelihood computation as *PredictEAM*, by giving an EAMC and iEAM. We illustrate its computation process in Figure 5. We revisit the MoE model from Figure 1. After R2 finishes dispatching the token to E[2,1], we need to initiate an online prediction. For this, MOE-INFINITY utilizes iEAM that traces the numbers of tokens passing through different experts in the current iteration. This iEAM is matched with prior EAMs in the EAMC (shown in ❶). Several matched EAMs might be returned. In such a case, we aggregate them and compute activation probability for each expert possibly to activate (shown in ❷). In this aggregation step, formally, the cell of each matched EAM is summed up and normalized on each row. To ensure future experts in proximity to the current layers can be prioritized, the layer proximity step (shown in ❸) adjusts the value in each cell through the formula $(1 - (i - l)/L)$, where $l$ is the current layer ID and $i$ is the future layer ID.

### 4.6. Cache optimizations

We implement several key optimizations for the expert cache:

**Enhancing the cache with prefetching.** To further improve performance, we integrate prefetching into the expert cache mechanism. Given the sequential nature of MoE model execution, where layers are processed in order, we can leverage the pEAM to predict the experts that are likely to be activated for the next layer. By prefetching experts into the cache, we reduce the likelihood of GPU stalls caused by on-demand expert fetching.

**Enhancing the cache with expert location information.** When deciding which expert to replace, we also consider the observation that: the initial layers of MoE models, which typically benefit less from prefetching due to less confident prediction of the group activation pattern at the start. By assigning higher caching priorities to experts in these initial layers, we not only counteract potential prefetching failures but also exploit the layer-by-layer execution property of MoE models: the subsequent layers are executed later and they are more likely to benefit from prefetching and thus less need caching.

### 4.7. Sparsity-aware expert cache algorithm

Finally, we can formally define the algorithm that realizes the sparsity-aware expert cache. Algorithm 1 presents the expert cache retrieval procedure. We collaborate expert cache with on-demand fetching for a conventional cache put procedure (steps 1-7). When the cache reaches its maximum capacity, an eviction mechanism is triggered to replace the

**Algorithm 1** Expert Cache Retrieval

**Require:** cur_EAM – Current iteration-level EAM, id
    – Requested expert ID, eamc – List of historical
    rEAMs, cache – Dictionary storing cached experts,
    cache_size – Maximum allowed cache size, m –
    Model instance with $L$ layers.
    **Output:** expert – Retrieved expert instance.
1: **if** id $\in$ cache **then**
2:    **return** cache[id]
3: **end if**
4: **if** |cache| < cache_size **then**
5:    cache[id] $\leftarrow$ FetchOnDemand(id)
6:    **return** cache[id] {Cache not full}
7: **end if**
8: p_eam $\leftarrow$ PredictEAM(eam, cur_EAM)
9: evict_expert $\leftarrow$ None, p_min $\leftarrow \infty$
10: **for** id, e **in** cache **do**
11:    n_token $\leftarrow \sum$ p_eam[e.layer_idx]
12:    p $\leftarrow \frac{(\text{p\_eam[e.layer\_idx]}+\epsilon)\cdot(1-\frac{\text{e.layer\_idx}}{L})}{\text{n\_token}}$
13:    **if** p < p_min **then**
14:      p_min $\leftarrow$ p, evict_expert $\leftarrow$ e
15:    **end if**
16: **end for**
17: delete cache[evict_expert.id]
18: cache[id] $\leftarrow$ FetchOnDemand(id)
19: **return** cache[id]

least relevant expert using prediction. The intuition is to
find the expert that has the least likelihood to be reused in
future iterations. As shown in Section 4.5, we compute the
likelihood for each expert by identifying the most similar
historical EAM (steps 8). The expert matched guarantees
similar overall activation pattern. We then computes a prior-
ity score, with layer decay taken into account (steps 9-16).
As expert from all layers needs to be considered, the decay
starts from the first layer. Finally, The expert with the lowest
priority is removed from the cache, making space for the
new expert that is fetched on demand, added to cache and
returned. (steps 17-19). The prefetching mechanism can be
integrated into the FetchOnDemand function. We omit
the detailed implementation here for brevity.

We also provide an example to understand how sparsity-
awareness helps make better cache performance than con-
ventional LRU (implemented in most inference systems,
such as vLLM and Ollama) and statistical counting ap-
proaches such as BrainStorm, as depicted in Figure 6. In
the second decoding iteration, an MoE model completes
the first layer and proceeds to the second. Once a token
is dispatched to $E[2, 1]$, Augmenting dependency-based
prefetching with an LRU cache, as in DeepSeek-Inference,
prefetching $E[2, 1]$ evicts $E[3, 1]$, leading to a buffer miss
when tokens route to $E[2, 2]$ (see (a) left). For statistical
counting approaches, as in BrainStorm, uniform activation

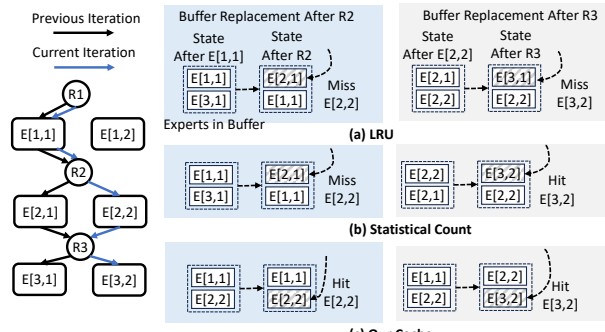

Figure 6: Example of integrating caching with prefetching.
LRU is the most commonly implemented technique in SOTA
systems such as vLLM, Ollama, DeepSpeed and Statistical
Count is implemented in BrainStorm.

means $E[1, 1]$ could route to $E[2, 1]$ or $E[2, 2]$, resulting in a
buffer miss (see (b) left). Our method, using a request-level
$EAM[[1, 2], [0, 3], [0, 3]]$, keeps $E[2, 2]$ from eviction, en-
suring a cache hit and better latency (see (c) left). When the
token enters layer 3, LRU method misses $E[3, 2]$, causing a
buffer miss (see (a) right). Statistical counting method iden-
tifies and prefetches the layer 3 expert but risks future misses
by evicting $E[2, 2]$ (see (b) right). Our strategy accurately
predicts and retains $E[2, 2]$ and prefetches preventing its
eviction and optimizing cache prioritization (see (c) right).

## 5. Evaluation

In the evaluation, we try to answer the following ques-
tions:

- Whether MOE-INFINITY achieves low latency under typ-
ical local deployment scenario?
- How MOE-INFINITY perform under long-context?
- Whether our prediction method is robust and efficient?

**Models.** We include popular open-sourced MoE models
in our evaluations including Google Switch Transform-
ers (Fedus et al., 2021) in the size of 30-100 GB depend-
ing on the configuration, DeepSeek-V2-Lite (DeepSeek-AI,
2024) (31GB), Meta NLLB-MoE (Costa-jussà et al., 2022)
(220GB), Mixtral-8x7B (Jiang et al., 2024) (120GB), and
Snowflake-Arctic (Snowflake AI Research) (900GB). For
these models, we report their results with the following con-
figurations if no further mentioned: DeepSeek-64x2.4B (de-
noted as DeepSeek), Switch-128x0.2B (denoted as Switch),
NLLB-128x0.4B (denoted as NLLB), Arctic-128x4B (de-
noted as Arctic), and Mixtral-8x7B (denoted as Mixtral).

**Datasets.** We used a large variety of LLM tasks (290 tasks
in total) contributed by three datasets to evaluate the perfor-
mance and robustness of MOE-INFINITY. These datasets in-
clude BIGBench (166 tasks), FLAN (66 tasks), and MMLU
(58 tasks). More specifically, these LLM tasks include
reasoning, contextual question answering, free response,

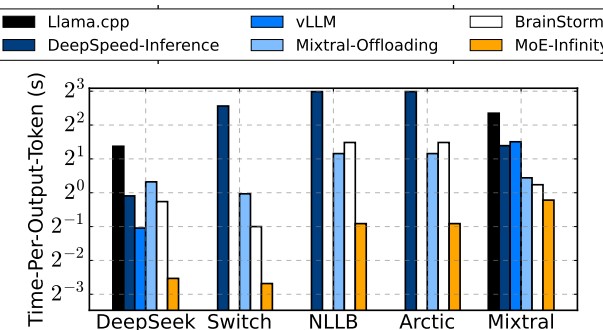

Figure 7: Decoding latency. vLLM and Llama.cpp does not support Switch, NLLB and Arctic, thus results are omitted.

translation and many more.

**Baselines.** We evaluate MOE-INFINITY against many SOTA baseline systems: (i) **DeepSpeed-Inference**, configured for optimized LLM inference (FastGen (Holmes et al., 2024)). DeepSpeed-Inference is the only mainstream LLM serving library that not only offers leading performance (comparable to vLLM and TensorRT-LLM) but also supports efficient offloading when GPU resources are limited. (ii) **Llama.cpp**, a high-performance inference engine optimized for environments with restricted GPU availability. By default, Llama.cpp stores all model parameters in CPUs and offloads computations to GPUs using highly optimized memory copy and caching kernels. (iii) **Mixtral-Offloading**, specialized in offloading-efficient MoE model inference, implements optimized parameter prefetching and caching strategies tailored for these models. (iv) **BrainStorm***, a leading dynamic neural network inference engine that implements model-level tracing to optimize operator scheduling, caching and prefetching. BrainStorm is not open-sourced and its design does not natively support MoE-based LLM tasks. We thus extended and implemented BrainStorm's model-level analysis in MOE-INFINITY.

**Hardware.** We show our experimental results with a single NVIDIA RTX A5000 GPU, connected to host memory via a dedicated PCIe 4.0 interface (32GB/s). We use 64GB memory for Switch, 256GB for NLLB, 32GB for DeepSeek, 128GB for Mixtral and 1TB for Arctic, as the model parameters need to be fully fitted into the Host memory.

### 5.1. End-to-end experiments

We now assess the performance and benefits when putting MOE-INFINITY in action for serving MoE models.

**End-to-end performance.** We report the end-to-end performance of MOE-INFINITY and baseline systems. Here, latency is reported as the time-per-output-token (decoding latency). We report the performance with a prompt length of 512 and a decoding length of 32. The latency is reported as the average of all datasets.

Figure 7 reports the end-to-end performance. For Mixtral—our worst-case scenario due to its small number of large-sized experts per layer and relatively high activation ratio—MOE-INFINITY achieves a latency as low as 836ms. Across all offloading-supported baselines, MOE-INFINITY demonstrates latency performance comparable to Brain-Storm and Mixtral-Offloading, with a 1.4× improvement. In contrast, vLLM and DeepSpeed-Inference exhibit higher latency, primarily due to their low cache hit rate. For offloaded parameters, Llama.cpp computes on the CPU, further contributing to performance degradation.

For MoE models with more experts per layer and lower selective activation ratios, the performance gains of MOE-INFINITY become more significant. For Switch, NLLB and DeepSeek, MOE-INFINITY achieves the 155ms, 531ms and 173ms latency, both numbers comparable to those with the model running fully in GPU. This means significant GPU saving by MOE-INFINITY: achieving similar latency performance, MOE-INFINITY requires a single GPU while the non-offloading alternatives requires 8 GPUs for NLLB and 4 GPUs for Switch. Other offloading-supported systems, however, cannot provide such a promise. BrainStorm and DeepSpeed-Inference suffers from inaccurate prefetching, creating extra traffic on PCIe link that blocks the on-demand fetching when needed.

For Arctic, the largest MoE model with 900GB of parameters, the model is composed of small-sized experts. MOE-INFINITY becomes the only available serving system that can offer competitive inference performance with a single GPU, vastly surpassing other baseline systems.

**Long context performance.** We evaluate the decoding performance of MOE-INFINITY and baseline systems under longer generation lengths, ranging from $2^{12}$ (4096) to $2^{17}$ (131072) tokens. This scenario frequently occurs in chain-of-thought (CoT) test-time scaling process (Wei et al., 2022). The LongBench dataset is used to provide meaningful prompts, ensuring continuous generation until the maximum length is reached. DeepSeek supports a maximum context length of 32K; beyond this limit, we enforce generation even after encountering EOS tokens. We stop at $2^{17}$ token since all systems OOMs.

Figure 8 reports the end-to-end performance. As the context length increases from $2^{12}$ to $2^{17}$, total computation time rises from 50ms to 160ms. Additional latency stems from offloading mechanisms. Increasing the context length leads to a larger KV-cache size in GPU memory, which in turn reduces the available buffer size for caching experts.

For MOE-INFINITY, the activated parameter size for DeepSeek-V2-Lite is 3GB per decoding iteration. Before a context length of $2^{15}$, the buffer size remains above 3GB, leaving sufficient space for caching. However, as the context

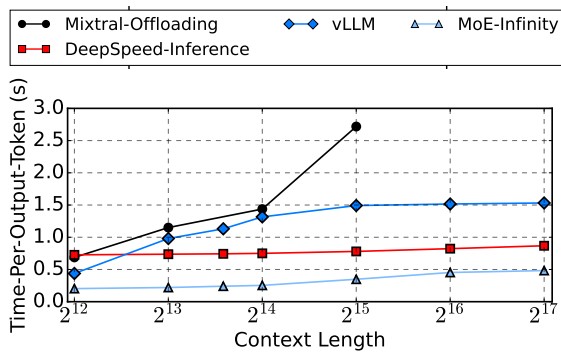

Figure 8: Decoding latency over long context. Using DeepSeek-V2-Lite with max context length 128K. We show top-4 systems in long context generation.

length increases, fewer experts can be cached—for example, at $2^{16}$, the buffer size shrinks to 2GB, and at $2^{17}$, it further decreases to 1GB—leading to performance degradation. Eventually, MOE-INFINITY resorts to on-demand fetching due to insufficient caching capacity. The on-demand fetching in MOE-INFINITY increases the latency by 137ms, being less than vLLM and Mixtral-Offloading. The reason is that MOE-INFINITY keeps all KV-cache in GPU memory, resulting in less fetching under long context.

As for vLLM, we observe that decoding latency increases more significantly as context length grows. vLLM also offloads KV-cache together with expert parameters, while the longer the context, the larger the KV-cache needs to be fetched to GPU at each layer. The KV-cache traffic causes contention with expert fetching, which delays and even blocks expert prefetching, leading to further performance degradation. DeepSeep-Inference is more stable in latency as it experience constant blocking time due to expert prefetching by index.

### 5.2. Micro-benchmark

Finally, we aim to evaluate the optimal parameter and robustness of the MOE-INFINITY activation tracer.

**EAMC Capacity.** Users of MOE-INFINITY may wonder how to determine the best EAMC capacity. To explore this, we adjusted the EAMC capacity while serving various MoE models. Figure 9 presents the results. We observed that increasing the capacity from 1 to 120 allows all MoE models to achieve their lowest average latency. Our findings highlight two key points: (i) A suitably small EAMC capacity (3% of the total number of requests), even amidst a challenging mixed LLM inference workload involving 290 tasks from three datasets, is adequate for capturing most MoE activation patterns, thus efficiently supporting MOE-INFINITY's prefetching and caching mechanisms, and (ii) The effectiveness of the EAMC capacity consistently manifests across different MoE models, indicating that the EAMC design can be generalized.

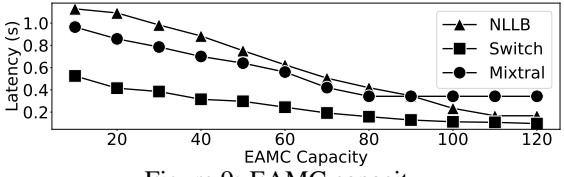

Figure 9: EAMC capacity.

| Workload Setup | NLLB | Mixtral | Arctic |
|---|---|---|---|
| MMLU tasks | 0-14-43 | 0-9-49 | 28-43-45 |
| BIGBench tasks | 1-15-49 | 0-11-49 | 0-29-42 |
| MMLU $\mapsto$ BIGBench | 0-6-17 | 0-27-42 | 2-28-41 |
| BIGBench $\mapsto$ MMLU | 3-11-24 | 0-35-46 | 5-27-45 |

Table 3: Handling workload changes. Numbers in each cell mean the minimum, mean and maximal numbers of requests required to recover low latency after a workload change.

**Robustness with workload changes.** Addressing concerns about handling workload changes, we tested MOE-INFINITY's tracer with task shifts, and measured the minimum, average, and maximum number of requests needed to restore low latency. The responsiveness to workload changes is shown in Table 3. In the first experimental group, we randomly shifted between LLM tasks within the same dataset. Experiments showed that within the same dataset, models returned to optimal latency after around 50 requests. Each task has 1000 input sequence on average, recovery from task shift needs 5% requests in the worst case. When switching between datasets (e.g., MMLU to BigBench), models adapted faster, averaging 30 requests for latency recovery. Each dataset samples 50K inputs, recovery from dataset shift needs less than 0.1% requests on average. This quicker adaptation is due to the reuse of activation patterns across similar tasks shared by these datasets, as highlighted in our trace study.

## 6. Conclusions

MOE-INFINITY is the first system to enable personal machines to achieve competitive performance when running large MoE models, such as the popular DeepSeek. Its key design, a sparsity-aware expert cache, has been extensively evaluated across various MoE models and LLM tasks, demonstrating 2.7–13.7× performance improvements over strong baselines. We anticipate that MOE-INFINITY will make it easier and more efficient for AI developers to deploy local MoE-based inference services. Additionally, its open-source nature is expected to drive rapid adoption and further innovation.

## Impact Statement

This paper presents work whose goal is to advance the field of Machine Learning. There are many potential societal consequences of our work, none which we feel must be specifically highlighted here.

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

## A. Existing Post-Hoc Predictors

Current post-hoc predictor (without changing model architectures and finetuning) does not fully capture the sparse activation properties of the MoE models.

**(1) Prediction based on dependency.** Predictors estimate expert activation based on memory dependency (Huang et al., 2020; HuggingFace, 2024; Aminabadi et al., 2022). As experts in one MoE layer all have memory dependency on the same router, such approaches fail to capture selective (S) and grouped (G) properties of sparse activation. Reuse (R) is not considered under the same scope.

**(2) Prediction based on counts.** Predictors use aggregated frequency counters on each expert to estimate activation (Cui et al., 2023; Jung et al., 2023). As experts tend to show uniform activation in the long run, this fails to capture the sparsity (S). In addition, individual counters cannot instruct the grouped activation (G) within and across layers.

**(3) Prediction based on locality.** Predictors estimate expert reuse based on heuristics such as LFU and LRU (Eliseev & Mazur, 2023; Jung et al., 2023; Cui et al., 2023; Aminabadi et al., 2022). Although only activated experts are considered (S,R), the reuse prediction is not applied across iterations, failing in the decoding phase.

## B. Practical Concerns

### B.1. Predictor Runtime Efficiency

We design EAMC to have fixed capacity, thereby limiting both memory costs and the time required to find a matching EAM. When the EAMC reaches its capacity, it necessitates the replacement of an entry within the collection. Our replacement strategy is guided by two main objectives: first, to record the most recent EAM, thereby quickly adapting to changes in workload; second, to maintain diversity within the recorded EAMs. Consequently, we opt to replace an EAM that is most similar to the incoming one. To implement this, we compare the new EAM against all existing ones in the EAMC, replacing the one that shows the shortest cosine distance. We illustrate the EAMC replacement process in Figure 10. Here, consider that the EAMC capacity is 3. Upon a new prompt P4 finished its trace, we compute the cosine distance between the EAM4 with all EAMs in the EAMC. The distances show that EAM4 is similar to EAM3, and we thus evict EAM3 and accommodate the EAM4.

**Runtime overhead.** The capacity of an EAMC must be appropriately small, as a large capacity can impede its practicality. MoE models benefit from a modest EAMC capacity for two main reasons: (i) After pre-training, MoE routers are optimized to create specialized expert groups for token processing, limiting the number of groups to ensure efficient

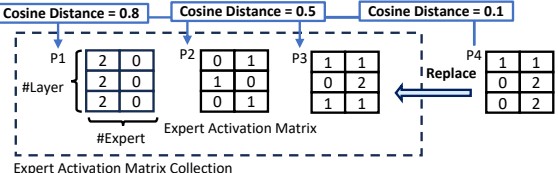

Figure 10: EAMC replacement example.

token dispatching and high accuracy, a characteristic underscored by leading research (Jiang et al., 2024; Team, 2024). (ii) Our evaluations in Section 5.2 indicate that a modest EAMC capacity, from hundreds to thousands, suffices for various LLM tasks and adapts well to task shifts, with the added advantage of negligible matching costs compared to model decoding latency. Searching for the most similar EAM is essentially a matrix multiplication on CPU (Douze et al., 2024). We measured the cost to be 21us per query under 1K EAMs and 226us for 10K EAMs. The frequency of the query is at most once per MoE layer for each (batched) input. Both memory and computation overhead are less than 1% of the model inference latency (typically >120ms per token).

**Capacity bound.** To analyse the upper bound of the number of cluster needed for a given cut-off distance under diverse inference requests, we formulate the EAMC construction as a sphere covering problem on the cosine distance, with each EAM as a vector in the space. In such a sphere space, arbitrary EAM can be projected to a point on the sphere, as under cosine distance, the EAM is normalized to unit vector. If we can find the minimal amount of the cluster that covers all area of the sphere, then the centroids of the clusters can be the representative EAMs for any given sequence. Theorems (Rankin, 1947; Dumer, 2007) shows that total number of cluster needed to cover the expert activation patterns is finite and polynomial complexity regarding the number of experts. In detail, this guarantees a lower bound of 75% cosine similarity by using $2LE$ EAMs and lower bound of 98% cosine similarity by using $\frac{1}{2}LE\ln(LE)$ EAMs. We observe that $E$ of SOTA MoE models ranges from 8 to 128 and $L$ ranges from 24 to 64 (Fedus et al., 2021; Jiang et al., 2024), leading to 40K EAMs with 160MB memory.

**Runtime optimizations.** The high computational complexity of the clustering algorithm makes this enhancement difficult to deploy. Consider the case of serving Arctic-128x4B for the FLAN dataset (which includes 66 LLM tasks). The clustering algorithm needs to handle over 1 million EAMs, each forming a 4480-dimensional vector (the flattened EAM). To our knowledge, no existing clustering libraries (e.g., FAISS (Douze et al., 2024)) can efficiently handle this workload. Hence, we adhered to the above simple but effective design for EAMC and left its enhancement with clustering algorithms for future work.

## B.2. System Implementation

**Support multiple GPUs.** We implement expert parallelism to support the use of multiple GPUs on a server. Concretely, we use a hashing function to assign the experts to different GPUs based on their IDs. All experts are kept in the host DRAM. While executing the MoE layers by layers, we use this hashing function to know which GPU is going to accommodate an expert needed for prefetching or execution. When the GPU is spread across multiple NUMA nodes, we will pre-partition the experts based on NUMA nodes, ensuring that these experts are only assigned to the GPUs in the designated NUMA node.

For each GPU, we create an independent I/O thread to manage the prefetching and caching. This thread uses pinned memory and DMA operations to optimize data transfers between the GPU and host DRAM, and a single thread is sufficient to saturate the bandwidth provided by PCIe 4.0 (32GB/s). For higher PCIe versions, we support creating multiple such threads per GPU.

For now, most open-source MoE models can be fitted into the host memory (up to 1TB) of a commodity multi-GPU server. We leave the multi-server support for future work.

**Memory management.** Given a MoE checkpoint, we keep its dense parts within the GPUs and turn on the offloading for its experts. This design is sufficient since the proportion of the experts' parameters comprising of 90-99% of the total parameters. For initializing the kv-cache, we will reserve the amount of GPU memory in corresponding to the maximal output length we observed in the open LLM datasets.

**Inference runtime integration.** We have integrated the above prefetching and caching mechanisms into PyTorch and support numerous kernel optimization, such as FlashAttention. Our current inference runtime supports checkpoints in PyTorch formats and HuggingFace formats.

**Failure recovery.** MoE-INFINITY can checkpoint its EAMC together with the MoE checkpoints. Once recovered from the failure, it reloads the EAMC to efficiently resume its prefetching and caching performance.

