# OpenReview forum: "MoE-Infinity: Efficient MoE Inference on Personal Machines with Sparsity-Aware Expert Cache"
_ICML.cc/2025/Conference — Submitted to ICML 2025_

### Official Review · Reviewer_Wuz9 · 2025-03-08

**Overall Recommendation:** 3

**Summary:**

The authors focus on the problem of high latency in MoE inference on personal machines with limited GPU memory. They observe that most existing offloading-based inference systems fail to effectively utilize the sparsity of expert activations during inference, leading to poor cache performance and high latency. It is interesting to develop the sparsity-aware expert cache to trace the sparse activation of experts and guide the replacement and prefetching of the experts. The evaluation shows the throughput improvements in per-token latency compared to the SOTA systems.

**Claims And Evidence:**

Yes. The claims are easy to follow and the evidence is clearly supported.

**Essential References Not Discussed:**

No. I think the references are adequately covered.

**Ethical Review Concerns:**

N/A.

**Experimental Designs Or Analyses:**

Yes. The experiments are correctly configured and the insights obtained from the experiments are clearly explained.

**Methods And Evaluation Criteria:**

Yes. The authors use typical open-source MoE models and LLM tasks.

**Other Comments Or Suggestions:**

Overall, I like this paper and the technical depth is fine in most aspects.

**Other Strengths And Weaknesses:**

The authors leverage the sparsity of expert activations during inference to effectively manage the expert cache. This analysis helps the readers optimize the decoding procedure and deploy efficient MoE inference services.

**Questions For Authors:**

The authors propose a method to predict expert activations during decoding based on historical activation traces. Could you please give more details on how to manage the cache at the system level, especially when data needs to be transferred between GPUs and CPUs?

**Relation To Broader Scientific Literature:**

This paper is strongly related to the MoE model design and deployment.

**Theoretical Claims:**

Yes. The problem of activation prediction in the decoding stage is correctly formulated.

---

> ### Author Rebuttal · Authors · 2025-04-01
>
> Thank you for your valuable feedback. We would like to address your comments and questions as follows.
>
> # Questions For Authors:
>
> ## 1. The authors propose a method to predict expert activations during decoding based on historical activation traces. Could you please give more details on how to manage the cache at the system level, especially when data needs to be transferred between GPUs and CPUs?
>
>    We manage the expert cache at the granularity of individual experts, with each expert's parameters stored as a contiguous memory block. Our system incorporates several techniques to ensure efficient CPU–GPU transfers:
>
> - **Avoiding repetitive expert fetching:** Once token-to-expert routing is determined at each layer, the experts that are already resident in GPU memory are "locked" to prevent eviction until their inference tasks are completed. While common frameworks such as Hugging Face launch selected experts strictly in ascending index order (e.g., executing expert 3, then 4, then 5 if those are selected), MoE-Infinity prioritizes experts already in cache, enabling faster inference by overlapping their execution with the fetching of cache-missed experts.
> - **Non-blocking Eviction:** When an expert is evicted from the GPU cache, its corresponding parameters remain in CPU memory. This design choice eliminates the need for blocking transfers back to the CPU.
> - **Memory Pooling and Reuse:** Separate expert memory pools are maintained on both CPU and GPU. For the GPU pool, evicted experts do not immediately trigger memory deallocation; instead, their slots are simply marked as available. This avoids the unpredictable and sometimes costly latency (up to tens of milliseconds) associated with memory free operations.
> - **Optimized Data Transfers:** CPU–GPU data transfers use CUDA’s memory management APIs (e.g., cudaMemcpy) and are accelerated through pinned memory to enable direct memory access (DMA), reducing transfer latency.
>
> # Others:
>
> N/A

---

### Official Review · Reviewer_HHYU · 2025-03-09

**Overall Recommendation:** 2

**Summary:**

Mixture-of-Experts (MoE)-based Large Language Models have recently exhibited strong performance across a wide range of tasks. However, their substantial model size poses significant challenges for deployment in resource-constrained environments. Expert-based caching has emerged as a promising approach to alleviate memory constraints. In this work, the authors propose MoE-Infinity, a predictive strategy aimed at enhancing expert caching prefetching. They further develop a software package that supports multiple MoE architectures. Experimental results demonstrate that their implementation achieves superior TPOT (throughput per token) compared to the selected baseline, indicating its potential for more efficient MoE inference.

**Claims And Evidence:**

The author provides three major contributions in this paper.

Contribution 1 asserts that caching can be leveraged even when the batch size is set to 1. While the claim is supported by empirical traces collected by the author, it lacks novelty. This observation has been well-documented in prior work, including Mixtral-Offload [1], as well as in numerous earlier studies [2-4], to name a few. At this stage, expert locality—particularly during the decoding phase—should be regarded as a well-established fact rather than a novel insight.

Contribution 2 introduces a statistical approach to modeling the probability of expert reuse, with the corresponding algorithm detailed in Algorithm 1. While the paper describes the methodology, further discussion on its distinct advantages over existing approaches would strengthen this contribution.

Contribution 3 states that the proposed method achieves superior performance compared to multiple baselines presented in the paper. The author provides a thorough implementation in the assets and experimental evaluation, which offer sufficient evidence to substantiate this claim.

[1] Eliseev, Artyom, and Denis Mazur. "Fast inference of mixture-of-experts language models with offloading." arXiv preprint arXiv:2312.17238 (2023).

[2] Huang, Haiyang, et al. "Towards moe deployment: Mitigating inefficiencies in mixture-of-expert (moe) inference." arXiv preprint arXiv:2303.06182 (2023).

[3] Yi, Rongjie, et al. "Edgemoe: Fast on-device inference of moe-based large language models." arXiv preprint arXiv:2308.14352 (2023).

[4] Kong, Rui, et al. "SwapMoE: Serving Off-the-shelf MoE-based Large Language Models with Tunable Memory Budget." arXiv preprint arXiv:2308.15030 (2023).

**Essential References Not Discussed:**

See claims and evidence for a (sub)set of literature not discussed.

**Experimental Designs Or Analyses:**

This paper does not explicitly evaluate the core contribution—namely, how the proposed algorithm improves the cache miss rate of the sparsity cache compared to existing approaches. Without such a comparison, it is difficult to determine whether the observed TPOT improvements stem from the inherent advantages of the proposed method or are merely the result of engineering optimizations.

In addition a detailed latency breakdown across different benchmarks could significantly strengthen the experimental section. Profiling the contributions of individual components to overall latency would provide deeper insights into the sources of performance gains and further substantiate the effectiveness of the proposed approach.

**Methods And Evaluation Criteria:**

The author evaluates decoding latency, measured in TPOT, using a single-GPU scenario benchmark, which makes sense.

**Other Comments Or Suggestions:**

For Figure 6, it might be better to put the in-figure caption ((a), (b), (c)) to the top of each subfigure. I got a little bit confused when I first saw the figure.

**Other Strengths And Weaknesses:**

The paper is well written and I barely find any typos or major errors.

**Questions For Authors:**

I have some concerns regarding the vLLM MoE Offloading implementation referenced in this work. As far as I am aware, vLLM does not currently support expert-based caching mechanisms, and its CPU offloading can move non-expert parameters within the same layer as well (vllm/model_executor/models/utils.py#L487) -- the author also mentioned this in the long context experiment (line 409). This raises a potential issue: in a single MoE layer, the last expert may have been evicted in previous decoding iterations, which could make it an unsuitable example for evaluation, as it's not properly performing expert caching. Clarification on this point would be beneficial to ensure the experimental setup is an apple-to-apple comparison.

Additionally, I am curious about the author's decision to restrict the study to bsz = 1. While single-request scenarios may be relevant for personal assistant use cases, handling multiple requests concurrently is crucial for improving response quality—particularly in techniques such as beam search or Tree of Thoughts reasoning. Expanding the analysis to include multi-request scenarios could provide stronger empirical validation of the proposed method's practical applicability.

**Relation To Broader Scientific Literature:**

Expert-based caching and offloading mechanisms have been extensively studied in the literature. This work presents an extension specifically tailored for the decoding phase with a single request. However, given the current experimental design and analysis, the contribution appears to be relatively incremental.

**Theoretical Claims:**

I didn't find any theoretical claim in this paper.

---

> ### Author Rebuttal · Authors · 2025-04-01
>
> Thank you for the detailed feedback—we’ll revise the figures and writing for clarity, and address key concerns below.
>
> # Questions For Authors:
>
> ## 1. Clarification to vLLM MoE Offloading implementation to ensure apple-to-apple comparison.
>    Thanks for raising this point. We include vLLM as it is a widely used SOTA LLM inference engine that supports CPU offloading, and many reviewers and users requested its inclusion in our evaluation.
>
>    We are aware that vLLM achieves efficient on-demand parameter fetching, sometimes outperforming baselines with simple expert prefetching (such as Mixtral-Offloading), which can waste expensive PCIe bandwidth by prefetching unused experts. This is shown in Table 1, and we will clarify it in the revised version.
>    In fact, sparsity-aware expert caching (MoE-Infinity) is conceptually complementary to vLLM’s offloading engine, and we are investigating how to integrate MoE-Infinity into vLLM to further improve its performance on MoE models.
>
> ## 2. The reasoning behind the decision to restrict bsz = 1. Expanding the analysis to include multi-request scenarios.
>
>    In practice, the beam width used in models is relatively small (typically ≤ 4), and batch sizes in ToT-style reasoning (e.g., Tree of Thoughts, NIPS 2023) is 5. The result of batch size (1-32) is shown in **Reviewer S1MY Others Section Q3**.
>
> # Others:
>
> ## 1. Contribution 1 asserts that caching can be leveraged even when the batch size is set to 1. While the claim is supported by empirical traces collected by the author, it lacks novelty.
>
>   Our system, started in September 2022, was among the first to explore sparsity-aware expert caching. Since its open-source release, it has gained significant traction and has been continuously improved to deliver state-of-the-art MoE inference on memory-constrained machines.
>
>    Despite this, our contributions are fundamentally different from [1, 2, 3, 4], which also explains why our system significantly outperforms [1]—the only open-source library among them.
>    First, **Contribution 1** goes beyond highlighting expert locality (i.e., skewed expert reuse). It focuses on identifying ***when*** and ***how*** such reuse patterns can be robustly observed and exploited. Our core finding—distinct from [1–4]—is that skewed expert reuse emerges **only** at the **request level** during decoding, rather than across requests [3, 4], tokens [1], or tasks [2]. Capturing these patterns goes beyond frequency counting or Markov models; we use request-level tracing with continuous matching—an approach not explored and used in prior works [1–4].
>    Specifically:
> - **Mixtral-Offloading [1]** assumes a correlation between individual tokens and expert activation. However, in modern MoE-based LLMs using attention, token activation depends on context, so a token may activate different experts in different settings, making this correlation unreliable and thus resulting in poor performance in practice.
> - **[2]** suggests that task-specific inputs lead to activation skewness. However, in practice, as the number of prompts increases, expert usage tends to become uniform at the task level. With this uniformity, the expert cache cannot effectively prioritize which experts to retain, which explains why this method is not widely adopted in real-world deployments.
> - **EdgeMoE [3]** claims reuse patterns exist across requests, but this fails to hold when expert usage is aggregated, which trends toward uniformity. In fact, this was one of the early designs implemented in MoE-Infinity (as a parallel research work), and we soon realized that this claim does not work over long periods of LLM serving.
> - **SwapMoE [4]** assumes semantically similar consecutive prompts to maintain reuse patterns, making it hard to deploy in practice since this assumption does not robustly hold. In contrast, MoE-Infinity does not rely on this assumption and can robustly capture reuse patterns across diverse datasets and models.
>
> In our collaboration with a partner deploying DeepSeek-R1, none of the above methods were used due to poor performance; they’re now working with us to integrate MoE-Infinity in clusters potentially hosting thousands of GPUs.
>
> ## 2. While the paper describes the methodology, further discussion on its distinct advantages over existing approaches would strengthen this contribution.
>    We carry out further micro benchmarks to distinguish our approaches from baselines. The result of the cache hit rate can be found in **Reviewer S1MY Others Section Q4**, and the results of the ablation studies of MoE-Infinity system components can be found in **Reviewer HkeA Others Section Q2**.

---

### Official Review · Reviewer_HkeA · 2025-03-14

**Overall Recommendation:** 3

**Summary:**

The authors introduce MoE-Infinity, a system targeting efficient inference for MoE models with a batch size of one, designed for personal machines with limited GPU memory. MoE-Infinity dynamically traces sparse activation patterns of experts during inference and optimizes caching and prefetching decisions to minimize latency and memory bottlenecks. The method relies on an Expert Activation Matrix (EAM) to predict future expert activations based on past patterns and employs a sparsity-aware expert caching strategy. Two key optimizations—expert prefetching and incorporating expert location information—are introduced to enhance caching performance. MoE-Infinity achieves significant latency reductions (up to 13.7 $\times$) compared to state-of-the-art systems like DeepSpeed and vLLM.

**Claims And Evidence:**

The claims made in section 4.2 need to be improved as it is not sufficiently grounded in its current state. Details such as the type of dataset used to generate the figure, or the behavior of layers other than the last layer are not provided. See more details below.

**Essential References Not Discussed:**

None

**Experimental Designs Or Analyses:**

Yes, the experimental design is sound but could be improved (see suggestions in ablating the components of the proposed method below)

**Methods And Evaluation Criteria:**

The paper does a good job in comparing against other frameworks for efficient inference across multiple benchmarks.

**Other Comments Or Suggestions:**

None

**Other Strengths And Weaknesses:**

While the significance of the speedups achieved by MoE-Infinity and its potential impact on enabling efficient deployment of MoE models on personal machines is clear, I have several major concerns regarding the suitability of the paper in its current form for an ML audience:

- **Accessibility for General ML Audience:** The paper can be difficult to follow for a general ML audience due to its engineering-heavy focus. While the speedups are impressive, the ML novelty of the work is rather limited.
- **Clarity on Contributions to Speedup:** The paper does not clearly delineate which specific components contribute to the reported 13$\times$ improvement over other SotA systems. Are these gains primarily from optimized caching and prefetching, the enhanced eviction policy, or the expert location prior? The authors should provide detailed ablation studies to isolate the impact of each individual component.
- **Comparisons Across Frameworks:** While comparisons across various frameworks are valuable, the vastly different approaches and implementation details make it challenging to pinpoint what drives the improved speedup. A major emphasis is placed on the enhanced eviction policy, but this claim could be substantiated further. For example, within their exact pipeline, the authors could replace their eviction policy with simpler alternatives like Least Recently Used (LRU) for a lower bound of performance and Belady’s optimal solution for an upper bound of performance. This would clarify how much of the speedup arises specifically from their caching strategy versus broader engineering differences compared to other libraries.
- **Lack of Detail in Claims:** Several claims in the paper lack sufficient grounding or detail. For instance:
	- It is unclear what dataset was used to generate the histograms in Figure 2. If specialized data such as coding datasets were used, sparse patterns might emerge, whereas general datasets like WikiText may not show such specialization.
	- The histograms only show activation patterns for the last layer of Mixtral, which is less informative since routing decisions in initial and mid layers are typically less confident and more inconsistent in MoEs compared to the last layer. The claim made in Section 4.2 about activation patterns needs further validation across earlier layers.
	- The authors need to provide more precise explanations and sufficient details regarding these observations.

**Questions For Authors:**

None

**Relation To Broader Scientific Literature:**

The speedups achieved by MoE-Infinity and its potential impact on enabling efficient deployment of MoE models on personal machines is impactful.

**Theoretical Claims:**

None

---

> ### Author Rebuttal · Authors · 2025-04-01
>
> Thank you for your valuable comments and suggestions. We address your concerns and comments as follows.
>
> # Questions for Authors:
>
> No questions
>
> # Others:
>
> ## 1. **Accessibility for General ML Audience:** The paper can be difficult to follow for a general ML audience due to its engineering-heavy focus. While the speedups are impressive, the ML novelty of the work is rather limited.
>    Our paper targets the ICML “Machine Learning Systems” track, with a focus on system-level novelty. We intentionally avoid modifying ML algorithms or MoE architectures (i.e., ML novelty). Based on our interactions with the ML community, there is a strong preference for systems that serve unmodified full models, thus preserving model accuracy. This approach also ensures MoE-Infinity remains compatible with emerging ML techniques like multi-token prediction. We will revise the writing of our paper to suit the general ML audience.
> ## 2. **Clarity on Contributions to Speedup:** Are these gains primarily from optimized caching and prefetching, the enhanced eviction policy, or the expert location prior?
>
>    Thank you for this suggestion. We did have the ablation studies while designing MoE-Infinity to understand how much benefits each component contributed, but we could not include them in the submitted version due to the page limit.
>    The results of the ablation studies can be found in the table below. We disabled one component at a time and measured the resulting performance degradation.
>    The most significant degradation occurs when disabling expert-wise offloading, as MoE layers are inherently sparse—fewer than 25% of experts are activated per token per layer. Without fine-grained offloading, the system unnecessarily fetches inactive experts, leading to a 3-4x increase in latency.
>    Other system-level optimizations also play crucial roles. pinned memory and NUMA-aware placement affect PCIe transfer speed, which is critical in offloading-enabled inference with small batch sizes. Finally, the benefit of our caching strategy is strongly tied to the cache hit rate, which is directly impacted by expert access patterns and the eviction policy.
>
>
> | Ablation Components | DeepSeek-V2-Lite | Mixtral-8x7B |
> | :---: | :---: | :---: |
> | Complete MoE-Infinity | 0.181 | 0.867 |
> | w/o Expert Wise Offloading | 0.487 | 3.579 |
> | w/o Cache Strategy | 0.275 | 1.278 |
> | w/o PinMemory | 0.300 | 1.621 |
> | w/o NUMA | 0.228 | 1.222 |
>
> ## 3. **Lack of Detail in Claims:** Several claims in the paper lack sufficient grounding or detail.
>    Thank you for your comments. Figure 2 is based on 1,000 requests uniformly sampled from a mixed dataset comprising BigBench, MMLU, and FLAN. We observe the same phenomenon consistently across each individual dataset. These results indicate that state-of-the-art MoE models, such as DeepSeek and Mixtral, show limited or no clear evidence of task specialization.
>    We analyzed expert activation across MoE layers during long-context decoding and found that activation is both highly skewed within each layer and non-uniform across layers—the most frequently activated expert IDs differ from one layer to another.
> ## 4. **Comparisons Across Frameworks:** While comparisons across various frameworks are valuable, the vastly different approaches and implementation details make it challenging to pinpoint what drives the improved speedup.
>
> We have results comparing mispredict between MoE-Infinity and baselines: caching the top k% frequently activated experts (Top-K), LRU, and Belady.
> We present the results in the table below. MoE-Infinity achieves 11–34% higher hit rates than baselines. Notably, for models like DeepSeek and Arctic, which exhibit complex and sparse expert activation (i.e., 100s experts per layer and low expert selection), MoE-Infinity significantly outperforms all alternatives. Among baselines, Top-K performs better than LRU (by 3–7%) except DeepSeek.
> These results underscore the need for our designs: selecting representative traces and continuously predicting activation during decoding.
>
> |Model(slots)|Top-K|LRU|MoE-Infinity|Belady|
> |:-|:-:|:-:|:-:|:-:|
> |DeepSeek-V2-Lite-Chat(360)|22%|31%|42%|66%|
> |NLLB(52)|25%|18%|46%|52%|
> |Arctic(76)|12%|9%|43%|51%|
> |Mixtral(39)|5%|2%|20%|44%|
>
> We have also conducted a detailed breakdown analysis; please refer to [Others Section Q2](#2-clarity-on-contributions-to-speedup-are-these-gains-primarily-from-optimized-caching-and-prefetching-the-enhanced-eviction-policy-or-the-expert-location-prior) for further information.

---

> > ### Comment · Reviewer_HkeA · 2025-04-09
> >
> > The authors have addressed most of my concerns and I raise my rating accordingly. I highly recommend to address these comments in the main manuscript in case of acceptance.

---

### Official Review · Reviewer_S1MY · 2025-03-18

**Overall Recommendation:** 3

**Summary:**

The paper introduces MoE-Infinity, an inference system optimized for Mixture-of-Experts (MoE) models on personal machines with limited GPU memory. Driven by their finding of single batch inference exhibiting a high degree of activation sparsity, the authors design a sparsity-aware expert cache, which can trace the sparse activation of experts during inference. It develop an expert activation prediction method that traces the sparse activation of experts and carefully selects traces that can guide future predictions. While comparing MOE-INFINITY against several advanced inference system, the authors provide claim their proposed method can acheive 3.1–16.7× per-token latency improvements over numerous state-of-the-art systems, including vLLM, Ollama, DeepSpeed and BrainStorm across various MoE models.

**Claims And Evidence:**

Claim 1: Significant speedup in MoE generation latency. This claim is convincing from Table 1 and Figure 7. I also appreciate authors for experiments in figure 8 for long context.

Claim 2: Limited activation of experts during single request. This claim need more experimental validation. Figure 2 is not sufficient enough. Also the results are difficult to interpret. How does the expert activation varies across different layers of a MoE model? Can the authors provide a more detailed study of expert activation patterns (e.g. Normalized ratio of expert e1 from layer 1 getting activated) over the course of a long context decoding?

**Essential References Not Discussed:**

I think citing some relevant work on MoE compression (specifically expert dropping) and understanding efficiency gains associated with them could be effective.

**Experimental Designs Or Analyses:**

1. How does the performance of the proposed method works for batch size > 1 and how it compares with the existing baselines? It is important to fully appreciate the proposed method although I respect author's proposition of personal low-end GPUs.

2. Additional results reporting hit/miss rates for activation predictions is necessary. How often does MoE-Infinity mispredict which expert will be needed?

3. I am interested in a baseline which always keep only Top k% most activated experts based on a statistics estimated using a calibration set. Does the authors have some already conducted experiments related to this or can discuss why and how bad this could be?

**Methods And Evaluation Criteria:**

Yes. The proposed technique is novel and the innovation marks a significant advancement in making efficient LLMs more accessible on personal machines. The evaluation is also well thought and make sense for the problem at hand.

**Other Comments Or Suggestions:**

1. Carefully check typos: eg. ln 114: computaion -> computation.
2. Have the authors found that some experts are completely useless and negligibly activated specially in MoEs with 100s of experts? If yes, I think removing them permanently could further benefit the efficieny of proposed method.
3. It will be interesting to see how MoE-Infinity can combine with SoTA MoE compression techniques (e.g. low-rank, sparsity, etc) could further benefit the proposed method.

**Other Strengths And Weaknesses:**

The paper brings several interesting ideas related to efficient cache design in MoEs and evaluated across different models and baselines. Overall, I am inclined to increase my score further if some of my questions are address effectively during rebuttal.

Some weakness I would like to list which demand additional attention are:
1. MoE-Infinity is optimized for personal machines, its scalability to multi-GPU environments or cloud servers is not fully explored.
2. MoE-Infinity can adapts well to similar task, I am concerned about its performance under drastic or diverse task distributions.

**Questions For Authors:**

1. When the authors mention - "ln 137 MoE models with fewer experts (e.g., Mixtral), we observe only 25% activation per request." Do they find out that 75% of experts are never activated.

2. Does the expert activation behavior changes depending on the dataset used (e.g, C4 vs MATH dataset)? If yes, what are the author thoughts regarding its impact on the proposed technique?

**Relation To Broader Scientific Literature:**

Related work provide some relevant relation and differences with prior work.

**Theoretical Claims:**

No theoretical claims.

---

> ### Author Rebuttal · Authors · 2025-04-01
>
> Thank you for the helpful suggestions. Below, we address the remaining concerns.
> # Questions:
> ## 1. 75% of experts are never activated?
>
> We clarify this does not mean 75% of experts are never activated. The 25% activation rate is inherent to Mixtral’s design, activating the top 2 out of 8 experts per layer per token at each decode step. When tracking expert activation across many requests, all experts are utilized, with usage following a uniform distribution.
>
> ## 2. Expert activation changes depending on dataset?
>
> Yes. We explored this during the design of MoE-Infinity by mixing datasets and switching them during inference. Table 3 presents partial results.
>
> Our experiments show that MoE-Infinity is robust across datasets. When mixing numerous tasks from BigBench, tail decoding latency remains stable. In a more extreme test(MMLU->BigBench ->C4->MATH), we observed that latency briefly spikes when switching to BigBench but recovers within ~20 prompts. Latency stays stable from C4 to MATH, as BigBench already covers their activation patterns.
> Results are similar across models(DeepSeek, Switch, NLLB, Mixtral).
>
> ![image1](https://anonymous.4open.science/api/repo/ICML2025-CFD9/file/decoder_heatmap.png)
>
> # Others:
> ## 1. Expert activation varies across different layers?
>
> Expert activation is skewed within layers and non-uniform across layers, with top expert IDs varying by layer.
> The heatmap with layer-wise normalized activation count(Mixtral-8x7B on BIG-Bench, decode 128) shows that some layers focus activation on a small subset of experts, while others distribute activation more evenly. Similar patterns were observed in DeepSeek, Arctic across datasets like BigBench and SharedGPT.
>
> ![image2](https://anonymous.4open.science/api/repo/ICML2025-CFD9/file/shift_latency.png)
>
> ## 2. Multi-GPU not fully explored.
> MoE-Infinity supports multi-GPU via expert parallelism.
>
> The table below shows our multi-GPU results. Scaling to more GPUs lets MoE-Infinity cache more experts, greatly reducing latency esp. for large models like Mixtral-8x7B, NLLB, and DeepSeek-V2-Lite. DeepSeek’s 4/8GPU results omitted(fits in 2 GPUs).
>
> |GPU|Mixtral|NLLB|DeepSeek|
> |:-:|:-:|:-:|:-:|
> |1|0.867|0.167|0.226|
> |2|0.741|0.141|0.135|
> |4|0.337|0.137|X|
> |8|0.171|0.111|X|
>
> ## 3. Batch size > 1 with baselines?
>
> We are aware that small batch sizes >1(e.g., in Tree of Thought prompting) are relevant on personal machines, and MoE-Infinity supports such use cases.
> On GSM8K (prompt 512, decode 256), MoE-Infinity achieves 2–6× higher TPOT for batch sizes 1–32, with similar trends on other datasets. As batch size grows, reduced expert sparsity narrows the gap. At batch size 32 (already large for personal machines), MoE-Infinity outperforms most baselines and is only ~10% slower than vLLM, despite vLLM’s heavy kernel and engineering optimizations. This gap is expected to shrink as MoE-Infinity matures.
>
> |BS|DeepSpeed|vLLM|MoE-Inf|
> |:-:|--:|--:|--:|
> |1|0.76|0.49|0.18|
> |2|1.36|0.50|0.23|
> |4|1.92|0.55|0.34|
> |8|2.32|0.60|0.53|
> |16|2.82|0.68|0.78|
> |32|3.22|0.89|0.97|
>
> ## 4. How often does MoE-Infinity mispredict?
>
> We have results comparing mispredict between MoE-Infinity and baselines: caching the top k% frequently activated experts(Top-K), LRU, and Belady.
> We present the results in the table below. MoE-Infinity achieves 11–34% higher hit rates than baselines. Notably, for models like DeepSeek and Arctic, which exhibit complex and sparse expert activation(i.e., 100s experts per layer and low expert selection), MoE-Infinity significantly outperforms all alternatives. Among baselines, Top-K performs better than LRU(by 3–7%) except DeepSeek.
> These results underscore the need for our designs: selecting representative traces and continuously predicting activation during decoding.
>
> |Model(slots)|Top-K|LRU|MoE-Infinity|Belady|
> |:-|:-:|:-:|:-:|:-:|
> |DeepSeek-V2-Lite-Chat(360)|22%|31%|42%|66%|
> |NLLB(52)|25%|18%|46%|52%|
> |Arctic(76)|12%|9%|43%|51%|
> |Mixtral(39)|5%|2%|20%|44%|
>
> The above results highlight the importance of capturing request-level skewness using our advanced tracing and prediction mechanism, which selects representative activation patterns from past traces and continuously predicts expert usage as decoding progresses—a capability uniquely offered by MoE-Infinity among SOTA MoE systems.
>
> ## 5. Have the authors found that some experts are useless?
>
> When handling large many-task datasets(e.g., BigBench), all experts are eventually activated, with activation ratios converging toward uniformity over time. While some experts may be underused in short intervals,we do not prune them; instead, we trace these skewed patterns to guide future activation prediction, aiming for robust long-term serving.
>
> ## 6. Combine with compression techniques?
>
> MoE-Infinity is designed to serve unmodified models to preserve accuracy, with lossy optimizations considered complementary. Some users have integrated quantization methods like GPTQ, and MoE-Infinity still shows robust performance.

---

### Decision · Program_Chairs · 2025-05-01

**Decision:**

Reject

**Comment:**

This paper proposes an efficient MoE inference system designed for limited resource scenarios. In particular, it leverages the expert usage pattern to prefetch expert cache which results in 2.7–13.7× latency improvements.

There were various questions from the reviewers including the case of batch size greater than 1, multi-gpu scenario and mis-predict cases. The authors provided additional experiments to answer those questions. However, even with this discussion, the overall evaluation of the reviewers didn't move much to the strong positive side.

As a result, we have 3 weak accepts and 1 weak reject. In the reviewer discussion, a reviewer pointed that the paper is focusing on the system level implementation, but lacked some more details such as NUMA (only mentioned in appendix), Pin Memory, etc.

Overall, this is not satisfactory for ICML publication.